# In Situ Observation of Retained Austenite Transformation in Low-Carbon Micro-Alloyed Q&P Steels

Xiaoyu Ye [1,2], Haoqing Zheng [2], Gongting Zhang [3], Zhiyuan Chang [2], Zhiwang Zheng [2], Zhenyi Huang [1,*], Xiuhua Gao [4] and Guanqiao Su [2,*]

1. School of Metallurgical Engineering, Anhui University of Technology, Ma'anshan 243002, China
2. State Key Laboratory of Vanadium and Titanium Resources Comprehensive Utilization, Pangang Group Research Institute Co., Ltd., Panzhihua 617000, China
3. Pangang Group Co., Ltd., Panzhihua 617000, China
4. State Key Laboratory of Rolling and Automation, Northeastern University, Shenyang 110819, China
* Correspondence: huangzhenyi@ahut.edu.cn (Z.H.); suhangjoe@gmail.com (G.S.)

**Abstract:** Retained austenite (RA) transformation and its role in the mechanical properties of three low-carbon micro-alloyed quenching and partitioning (Q&P) steels was investigated utilizing in situ tensile tests and electron microscopy. Meanwhile, RA's strain-induced martensite transformation (SIMT) was analyzed and discussed in terms of the strengthening mechanism. The results show that the ductility of the Q&P steels relies on the size and morphology of RA. In addition, both affect RA's mechanical or thermostability. Dislocation density and carbon trapping should be considered in estimating the yield strength in the two-step Q&P process. V and Nb-Ti elements promote the formation of blocky RA. Ti accelerates the formation of film-like RA. For experimental Q&P steels with different processes and compositions, the true stress always keeps a linear relationship with the amount of transformed martensite, i.e., 30.38~46.37 MPa per vol. 1% transformed martensite, during the in situ tensile deformation.

**Keywords:** quenching and partitioning; retained austenite; strengthening mechanism; precipitation

## 1. Instruction

Quenching and partitioning (Q&P) steel, one of the third-generation advanced high-strength steels for lightweight automobiles, has been widely investigated due to its good combination of strength, toughness, and low cost [1–3]. The scientific principle for Q&P steel is that the supersaturated carbon in martensite can diffuse into austenite during the partitioning process, which results in the carbon-enriched retained austenite (RA) at room temperature. Thus, the excellent strength and ductility of Q&P steel are obtained by the effect of transformation-induced plasticity (TRIP). During the quenching process, the heat treatments consist of annealing in the temperature zone of full austenite (or two phases), and the quenching is performed in the temperature range of martensite-start ($M_s$) and martensite-finish ($M_f$). The partitioning stage is carried out at either a quenching temperature (one-step Q&P process) or a relatively higher one (two-step Q&P process) [4]. Currently, the one-step Q&P process is more suitable for continuous annealing lines in the industry.

Microalloying elements, such as Nb, V, and Ti, are traditionally added to the steel to improve the product of strength and ductility via the effect of grain refinement and precipitation strengthening. For TRIP steel, Perrard and Scott [5] found that excellent TRIP steel with a tensile strength of 980 MPa and a total elongation of above 20% could be obtained by adding V and N. Krizan et al. [6,7] investigated the effect of Ti, Nb, and Nb-V individually or in combination on the mechanical properties of C-Si-Mn-Al TRIP steels and pointed out that the addition of Ti, Nb, and Nb-V was a practical technology to significantly improve strength, with a slight degradation of the elongation. Our previous studies also

showed that V-alloyed TRIP steel shows a good combination of tensile strength of 1000 MPa and total elongation of 19% [8]. Similarly, Yan et al. [9] confirmed that Ti-alloyed Q&P steel shows a superior combination of strength and elongation, and the product of strength and elongation fills in the range of 19.6~20.9 GPa%. Zhang et al. [10] suggested that the Nb addition with a suitable amount increased the stability of RA by grain refinement in Q&P steel. At the same time, the volume fraction of RA decreased, which promoted the precipitation of NbC. However, the RA transformation and strengthening mechanism are unclear in the Q&P steels, where the vital carbide-forming elements Nb, V, and Ti are added.

The stability of RA, the critical index for evaluating the TRIP effect, plays a vital role in the mechanical properties of Q&P steel. Numerous studies show that the stability of RA relies on many factors, such as chemical composition, size, distribution, morphology, and surrounding phases [11–16]. In order to evaluate the stability of RA, many techniques, including the interrupted tensile test, XRD [17], saturation magnetization [18,19], in situ EBSD, and the tensile test [20], are widely used. Due to the limitation of the technology above, the RA transformation and strengthening mechanisms still need to be clarified in the Q&P steels, especially when the vital carbide-forming elements of Nb, V, and Ti are added. Recently, the X-ray stress apparatus (X-350), aided by a micro-electronic universal testing machine, has showed a significant advantage in the in situ analysis of RA since it provides a convenient way to track the volume fraction of RA during the tensile test [21]. In addition, many recent references have presented in situ observations of FCC phase structures in uniaxial tension [22–26]. However, they need to focus more on the RA stability of Q&P steel obtained by in situ testing methods.

The stability of RA in TRIP steel plays a crucial role in mechanical properties. There are limited papers to clarify the transformation mechanism of RA, although plenty of related studies have shown that the transformation of RA improves strength and ductility. Thus, in this study, the in situ tensile test and X-ray technology are used to investigate the stability of RA in the micro-alloyed low-carbon Q&P steels. Furthermore, the RA stability regarding the transformation and strengthening mechanisms in one- and two-step Q&P steels have been discussed regarding chemical composition, microstructure, and corresponding mechanical properties.

## 2. Experimental Material and Procedures

Chemical compositions of designed micro-alloyed steels, namely V-alloyed, Nb-Ti-alloyed, and Ti-alloyed steel, respectively, are listed in Table 1 (determined by mass spectrometry). The preparation process of the V-alloyed, Nb-Ti-alloyed, and Ti-alloyed steels is as follows. The cast ingots underwent a homogenization treatment under an argon atmosphere at 1250 °C for 3 h. After hot-forging, these billets were hot-rolled to the plates with a thickness of 3.5 mm. The beginning temperature of hot-rolling was 1060 °C, and the finishing temperature of hot-rolling was 890–910 °C. Subsequently, the steels cleaned with acid were cold-rolled to a 1.5 mm thickness. Then, the cold-rolled specimens with a dimension of 450 mm (perpendicular to the rolling direction) × 80 mm × 1.5 mm (length × width × thickness) were prepared for the Q&P process. The Q&P process was carried out by the continuous annealing monitor (MMS 300). These designed steels were subjected to different heat treatment regimes. The specimens were heated to 820 °C at a heating rate of 5 °C/s and held for 3 min for inter-critical annealing. Then, the specimens were slowly cooled to 760 °C at the cooling rate of 1 °C/s and further cooled to 380 °C at the cooling rate of ~30 °C/s for quenching and partitioning, and finally fast cooled to room temperature. The temperature of quenching and partitioning was slightly below the $M_s$ temperature (385 °C, 391 °C, and 382 °C in V-alloyed, Nb-Ti-alloyed, and Ti-alloyed steel, respectively), which was measured by the dilatometer analyses. For comparison, the Ti-alloyed samples underwent an immediate quench (first quenching) to 240 °C for 15 s, reheated to 380 °C, and kept for 6 min (the two-step Q&P heat treatment). The schematic map of one-step (black line) and two-step (blue line) heat treatments is shown in Figure 1.

**Table 1.** Chemical compositions of designed steels (wt.%).

| Steel | C | Si | Mn | Nb | Ti | V | Fe |
|---|---|---|---|---|---|---|---|
| V-alloyed | 0.2 | 1.7 | 2.0 | - | - | 0.1 | Bal. |
| Nb-Ti-alloyed | 0.2 | 1.7 | 2.0 | 0.02 | 0.03 | - | Bal. |
| Ti-alloyed | 0.2 | 1.7 | 2.3 | - | 0.02 | - | Bal. |

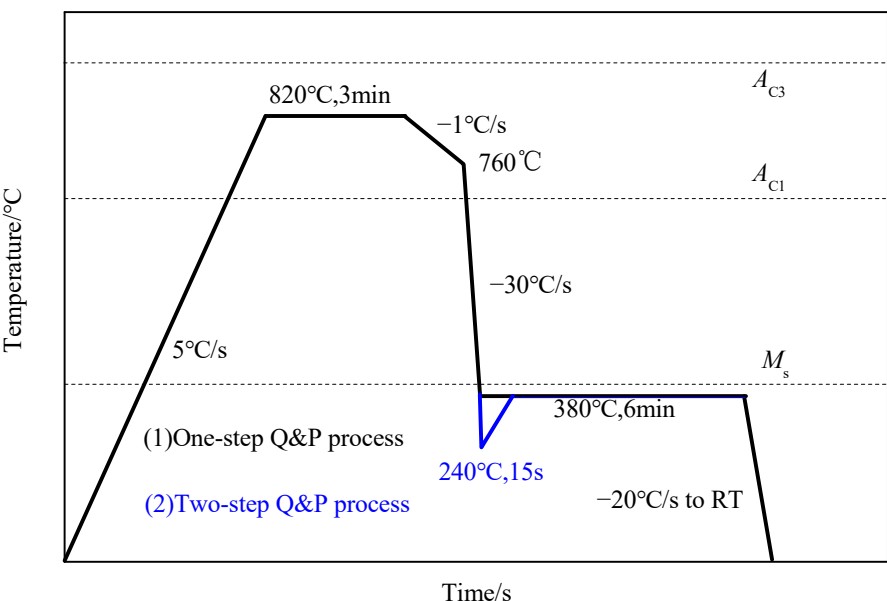

**Figure 1.** Schematic map of the quenching and partitioning (Q&P) process.

The prepared specimens were cut, ground, polished, and etched with 4% nitric acid. Then, the specimens were observed by a Zeiss Axio Observer A5 optical microscope (OM) (Jena, Germany) and a Zeiss Ultra 55 field-emission scanning electron microscope (FE-SEM) (Jena, Germany). The grain size of ferrite in the designed steels was calculated using a linear intercepted method after counting over 2000 grains. The volume fraction of ferrite was estimated by the ImageJ software. A JEOL-2100F field-emission transmission electron microscope (TEM) (Tokyo, Japan) was employed to reveal fine microstructure and precipitates. For TEM samples, 40 μm thin foils were prepared by a twin-jet electro-polishing technique using a solution consisting of 87.5% ethanol and 12.5% perchloric acid.

The volume fraction of RA was measured using a Rigaku D/max 2400 X-ray diffraction (XRD) (Rigaku, Tokyo, Japan) device with Co $K_\alpha$ radiation (λ = 0.1789 nm). The polished samples were scanned over a 2θ range from 0° to 120° with a step size of 0.02° and a scanning speed of 1°/min. XRD data were analyzed using the MDI Jade 5.0 software, and the integrated intensity of austenite peaks of (200), (220), and (311), and ferrite peaks of (200) and (211), were used to quantify the content of austenite using Equation (1):

$$V_i = \frac{1}{1 + G\left(I_\alpha / I_\gamma\right)} \tag{1}$$

where $V_i$ is the volume fraction of austenite, $I_\alpha$ and $I_\gamma$ are the integrated intensities of ferrite and austenite peaks, respectively, and the *G*-value is a parameter for each combination of $I_\alpha / I_\gamma$ [27,28].

The specimens with a cross-section of 20 × 1.5 mm$^2$ and a gauge length of 80 mm along the transverse direction for the tensile test were cut from the heat-treated samples. The tensile tests were performed at room temperature using an INSTRON 5569 machine (Norwood, MA, US) with a strain rate of 0.001 s$^{-1}$. The average values of three duplicated specimens for the individual test were used to ensure the consistency of the obtained

results. The tensile FE (finite element) models of specimens were established by using the FE software ABAQUS (Johnston, RI, US). Moreover, the work-hardening rate was calculated with the derivation of the true stress–true strain curve and discussed in terms of the corresponding microstructure. The fractured surfaces after the tensile test were observed using the Zeiss Ultra 55 FE-SEM, operated at 10 kV after ultrasonic cleaning with acetone. The changing volume fraction of RA against the tensile strain was measured using an X-ray stress apparatus (X-350) aided by a micro-electronic universal testing machine with a crosshead speed of 0.5 mm/min [21]. The dog-bone specimens with a section of $5 \times 1.5$ mm$^2$ and a total length of 60 mm were cut from heat-treated specimens. The length axis of the specimens was along the transverse direction. The volume fraction of RA was tracked with an interval of a certain strain during the tensile test. At the same time, in the present study, the stress values, i.e., 0, 100, 200, 300, 400, 500, 600, 700, 800, 900, and 1000 MPa, were applied to evaluate the volume fraction of RA using a similar method as mentioned above.

## 3. Results

### 3.1. Microstructure Morphology

The chemical composition and the Q&P process play an essential role in the microstructure evolution of steels. The microstructure of three designed steels subjected to a one- or two-step Q&P process is presented in Figure 2. Generally, ferrite is white in the OM figure and dark in the SEM figure, and martensite is black or brown in the OM figure and blocky with a white edge in the SEM figure. The martensite/austenite (M/A) constituents were presented as white islands in the OM figure and blocky with a white edge in the SEM figure. After the one-step Q&P process, the V-alloyed and Nb-Ti-alloyed steels (Figure 2a–d) consisted of ferrite (F), bainite (B), and martensite/austenite (M/A). The bainite marked by the yellow arrows is observed in Figure 2b,d. Compared with the one-step Q&P process, the Ti-alloyed specimen with a two-step Q&P process consisted of F and M/A, and bainite was not observed in Figure 2e,f. Figure 2 shows that the microstructure of the one-step Q&P process consisted of ferrite, bainite, and martensite/austenite island. The two-step Q&P process showed ferrite, martensite, and martensite/austenite island. Thus, the two-step Q&P process promoted the formation of martensite compared with the one-step Q&P process. The average grain sizes of ferrite in V-alloyed, Nb-Ti-alloyed, and Ti-alloyed steels were 6.51 μm, 8.37 μm, and 4.54 μm, respectively. This case indicates that a two-step Q&P process may refine the grain. Table 2 shows the volume fractions of each phase in the designed micro-alloyed Q&P steels. The main phase was martensite in all designed steels, i.e., over 61% in volume fraction. After the one-step Q&P process, the volume fractions of RA in V-alloyed steel and Nb-Ti-alloyed steel were 10.8% ± 0.7% and 12.90% ± 0.79%, respectively. The volume fraction of RA in two-step Ti-alloyed Q&P steel was 11.07% ± 0.51%. The result indicates that the amount of RA of the prepared specimen by the one-step Q&P process was similar to that of the two-step Q&P process. Since the temperature of quenching and partitioning was near the temperature of $M_\mathrm{s}$, the volume fraction of bainite in V-alloyed and Nb-Ti-alloyed steel was below 0.5%. Thus, the effect of bainite on mechanical properties can be ignored in the present study.

**Table 2.** The volume fractions of the different phases (%).

| Steel | Ferrite | Martensite | Retained Austenite |
|---|---|---|---|
| V-alloyed steel (one-step Q&P) | 17 ± 2.00 | 71 ± 2.00 | 10.8 ± 0.7 |
| Nb-Ti-alloyed steel (one-step Q&P) | 22 ± 2.00 | 64 ± 2.00 | 12.90 ± 0.79 |
| Ti-alloyed steel (two-step Q&P) | 27 ± 2.00 | 61 ± 2.00 | 11.07 ± 0.51 |

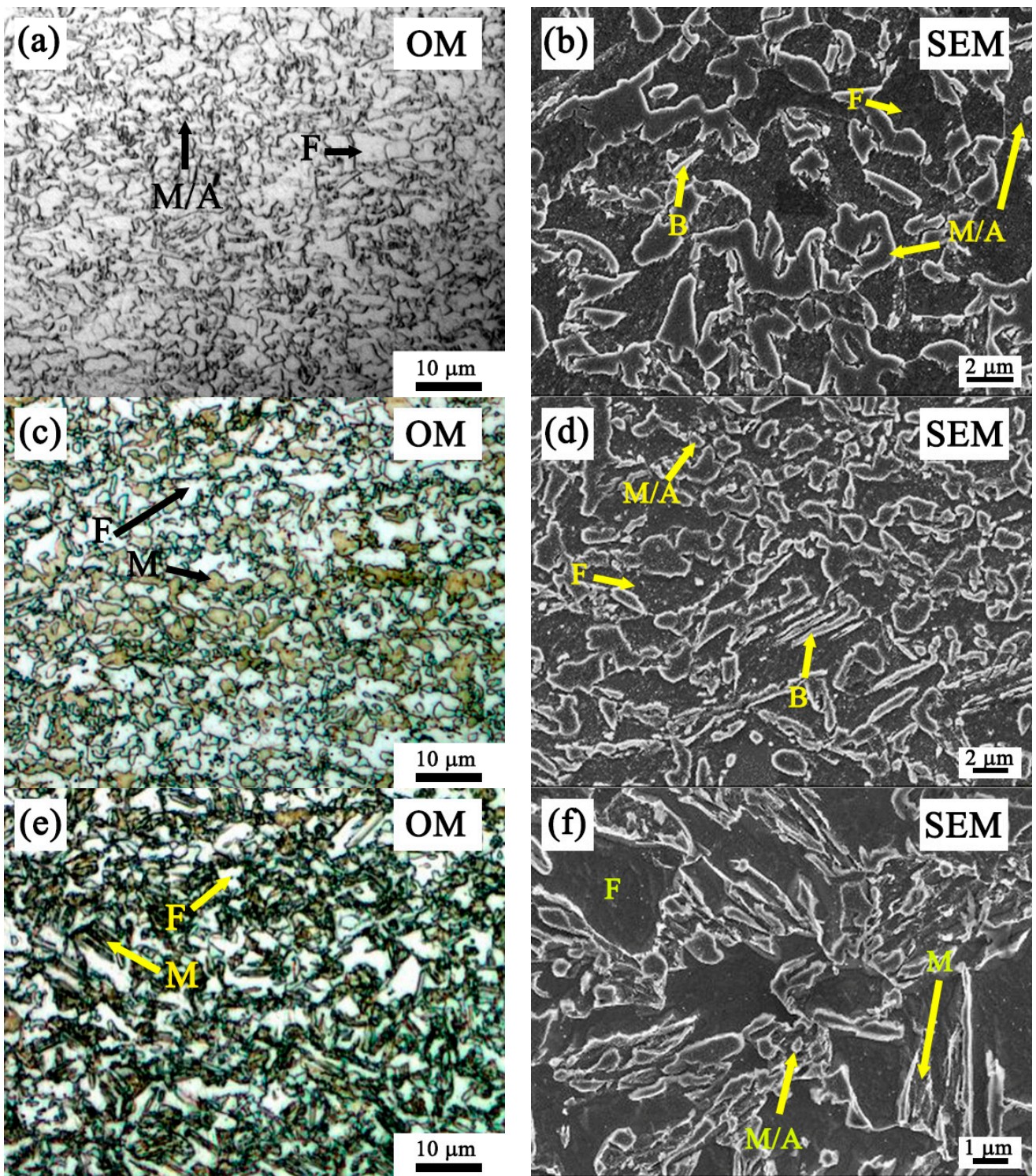

**Figure 2.** Microstructure morphology of the designed steels: (**a**,**b**) V-alloyed steel with one-step Q&P process, (**c**,**d**) Nb-Ti-alloyed steel with one-step Q&P process, and (**e**,**f**) Ti-alloyed steel with two-step Q&P process.

The morphology characteristics of RA and carbides are shown in Figure 3. A large amount of blocky RA was formed in the V-alloyed and Nb-Ti-alloyed steels after the one-step Q&P process (Figure 3a,b). Film-like RA was predominant in the Ti-alloyed steel after the two-step Q&P process (Figure 3c). The precipitated carbides were found in all designed V-alloyed, Nb-Ti-alloyed, and Ti-alloyed steels (Figure 3d–f). According to

selected area electron diffraction (SAED) patterns and EDS analysis results, VC, (Nb, Ti)C, and TiC were identified as the primary type of precipitation in V-alloyed, Nb-Ti-alloyed, and Ti-alloyed steels, respectively. The carbides of VC, (Nb, Ti)C, and TiC were precipitated from ferrite when the temperature was less than the starting precipitate temperature. The precipitate temperature of VC fell in the range of the holding temperature. Plenty of VC was formed in the V-alloyed steel. These VC particles may grow with a prolonged holding time. Thus, large VC particles were observed in V-alloyed steel (Figure 3d). The precipitation temperature of complex carbides (Nb, Ti)C was slightly higher at 800 °C. Thus, plenty of (Nb, Ti)C particles were also observed. The precipitate temperature of TiC was vastly higher than 830 °C. The observed TiC particles may be formed during the heating process of hot-rolling or solidification. These precipitated carbides were mainly distributed in ferrite. Thirty TEM photos of precipitated carbides were randomly counted. The average sizes of these precipitated carbides in V-alloyed, Nb-Ti-alloyed, and Ti-alloyed steels were $48 \pm 5$ nm, $10 \pm 3$ nm, and $20 \pm 4$ nm, respectively. The volume fractions of precipitated carbides in ferrite were 13.0%, 0.75%, and 0.67% for V-alloyed, Nb-Ti-alloyed, and Ti-alloyed steels, respectively. After considering the volume fraction of ferrite among all the phases, the volume fractions of precipitated carbides in V-alloyed, Nb-Ti-alloyed, and Ti-alloyed steels were estimated as 2.21%, 0.16%, and 0.19%, respectively. The decrease of carbides for the V-alloyed, Nb-Ti-alloyed, and Ti-alloyed steels ascribed the difference in the precipitated content. The relative atomic mass for V, Nb, and Ti was 50, 41, and 47, respectively. Assuming that the ideal precipitation behavior occurred, the ratio of the precipitated volume of VC, (Nb, Ti)C, and TiC was 16.12:5.08:3.90, which indicates that the volume fraction of carbides in V-alloyed steel should be the largest. Precipitation temperature is also an essential factor that influences the volume fraction. The precipitation temperature of VC ranged from 600 °C to 800 °C, which fell in the holding temperature and accelerated the precipitation process. The precipitation temperature of NbC was 800–1100 °C, and that of TiC surpassed 1100 °C. Thus, the heat treatment process in this study is not beneficial for the precipitation of NbC and TiC. Therefore, the volume of (Nb, Ti)C and TiC was lower than that of VC.

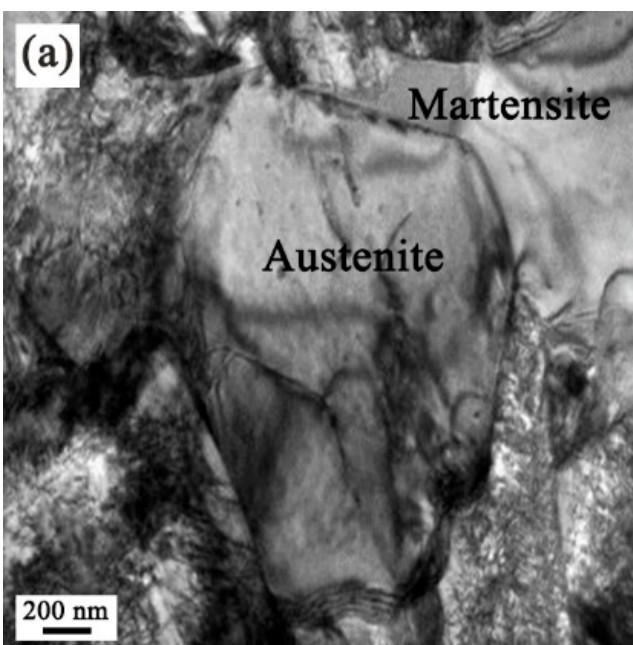
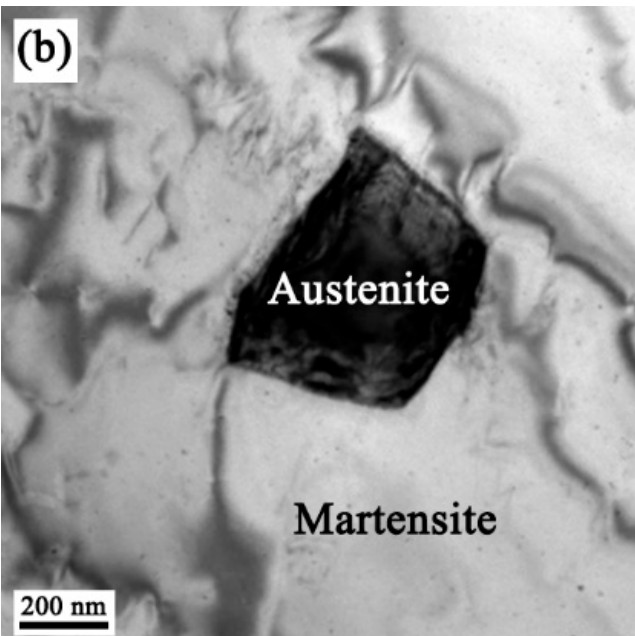

**Figure 3.** *Cont.*

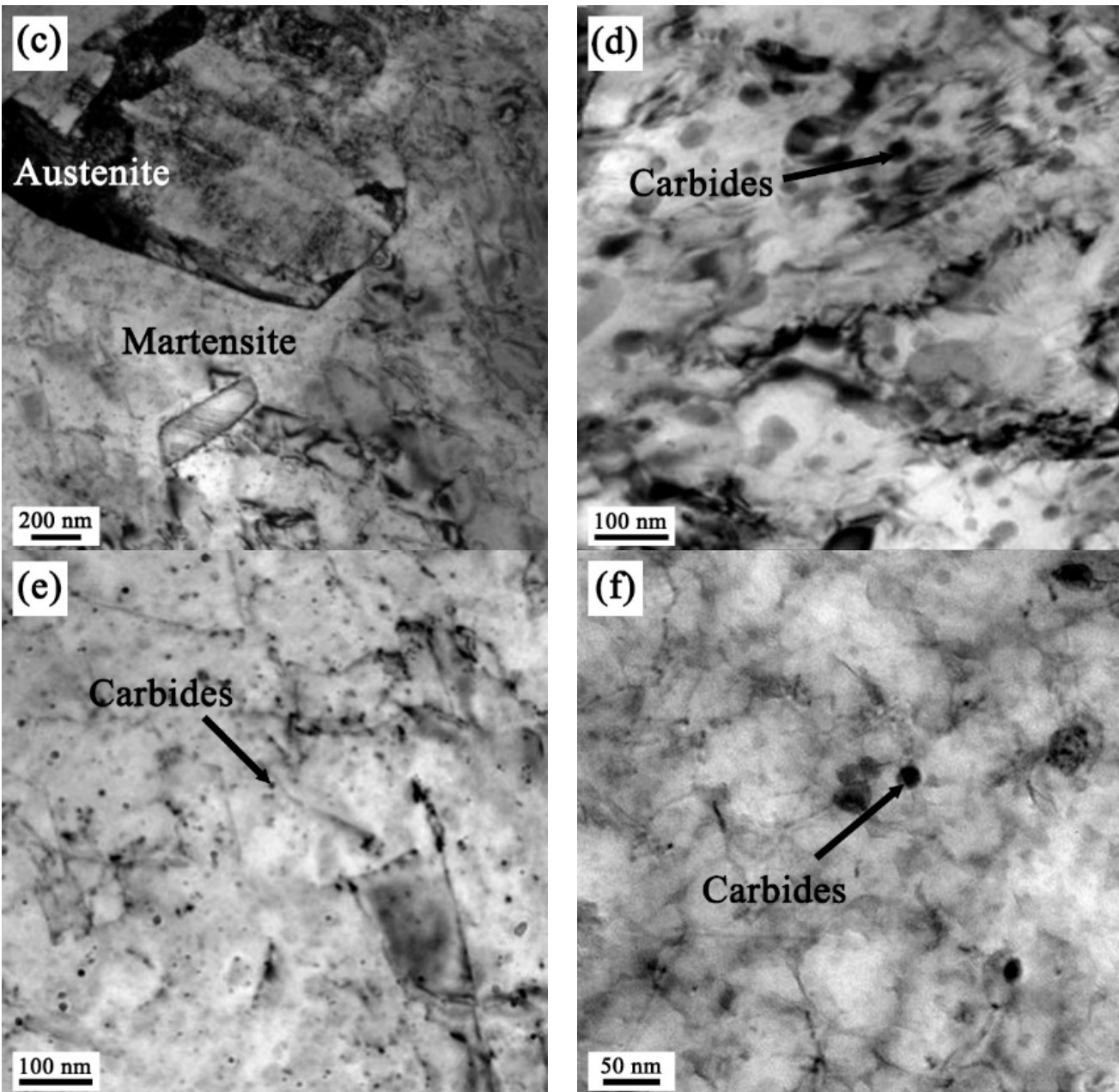

**Figure 3.** Morphology characteristics of retained austenite and carbides precipitated in the experimental steels: (**a**–**c**) RA in V-alloyed steel, Nb-Ti-alloyed steel, and Ti-alloyed steel, respectively. (**d**–**f**) Precipitation in V-alloyed steel, Nb-Ti-alloyed steel, and Ti-alloyed steel, respectively.

### 3.2. Retained Austenite and Its Transformation

The patterns of all designed steels prior to deformation and the evolution process of the calculated volume fraction of RA at different strains during the tensile test are illustrated in Figure 4. Figure 4a shows that all the prepared steels consisted of martensite (M) and austenite (A). During the tensile deformation process of all the prepared steels, the volume fraction of RA remained constant until the stress exceeded 500 MPa in all the studied steels, indicating that the tested steels bear elastic deformation. The RA content gradually decreased with the increasing strain during the plastic deformation process. Furthermore, under the same strain, the fraction of RA in Nb-Ti-alloyed steel was slightly higher than that of Ti-alloyed steel and V-alloyed steel. This case may ascribe to the higher initial volume fraction and the higher plastic strain of Nb-Ti-alloyed steel,

which indicates that Nb-Ti-alloyed steel showed the best stability of RA among these three micro-alloyed steels.

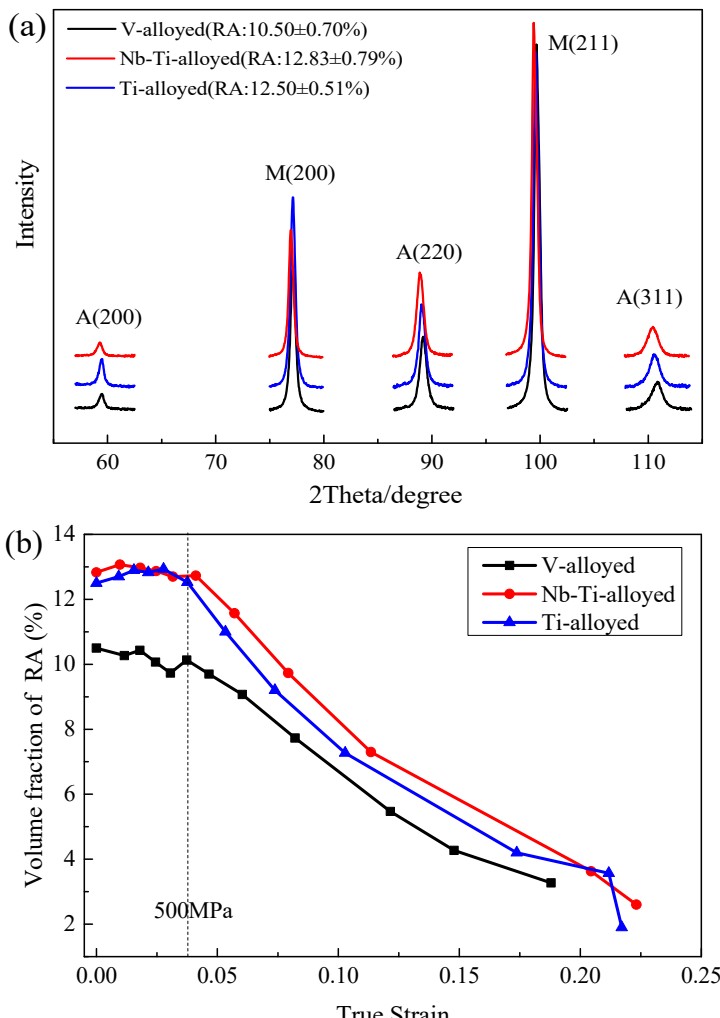

**Figure 4.** XRD diffraction patterns and corresponding volume fraction of RA in the investigated steel (**a**), and the volume fraction of RA at different strains (**b**).

### 3.3. Mechanical Properties

The mechanical properties of the designed steels are listed in Table 3. Their engineering stress–strain curves and corresponding work-hardening rate are presented in Figure 5. The tensile strength of all prepared steels surpassed 1000 MPa, and the product of tensile strength and total elongation was in the range of 17.84 GPa·% to 24.12 GPa·%, which shows a better combination of strength and elongation than that of Nb-V TRIP and Ti Q&P steel [7–9]. Moreover, the work-hardening rate of V-alloyed steel was higher than that of Nb-Ti-alloyed and Ti-alloyed steels within the true strain range of 0.02 to 0.10. The work-hardening behavior in Nb-Ti-alloyed and Ti-alloyed steels was similar. This phenomenon indicates that the one-step Q&P process could obtain the same mechanical properties carried out by the two-step Q&P process. The work-hardening rate was divided into three stages to clarify the different deformation stages more profoundly, as illustrated in Figure 5b. More details can be found in Section 4.2. Figure 2 shows that the microstructure of Ti-alloyed steel was bainite, martensite, and martensite/austenite island, while the microstructure of V-alloyed and Nb-Ti-alloyed steels consisted of bainite, ferrite, and martensite/austenite island. For iron and steel, the strength of martensite is high, and the strength of ferrite is

low. Thus, the yield strength of Ti-alloyed steel was more significant than that of V-alloyed and Nb-Ti-alloyed steels (Table 3).

**Table 3.** The mechanical properties of the prepared steels.

| Steel | Yield Stress (YS, MPa) | Tensile Stress (TS, MPa) | Total Elongation A (%) | TS × A (Gpa·%) |
|---|---|---|---|---|
| V-alloyed (one-step Q&P) | 668 ± 5 | 1115 ± 2 | 16.0 ± 0.5 | 17.84 |
| Nb-Ti-alloyed (one-step Q&P) | 622 ± 5 | 1005 ± 2 | 24.0 ± 0.5 | 24.12 |
| Ti-alloyed (two-step Q&P) | 718 ± 5 | 1058 ± 2 | 20.5 ± 0.5 | 21.69 |

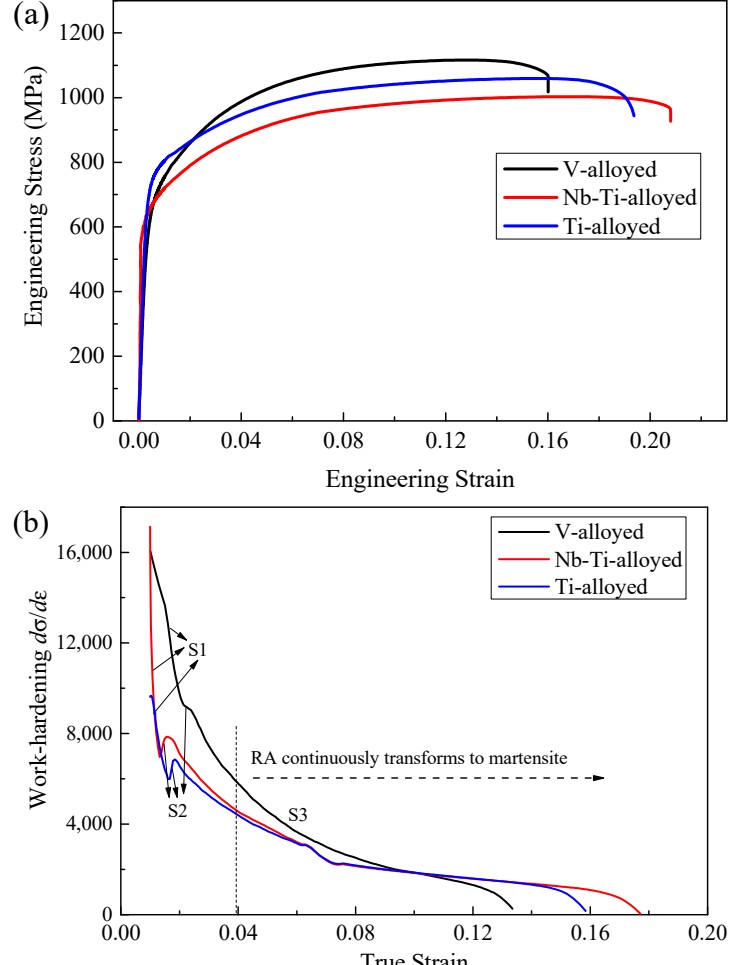

**Figure 5.** The engineering stress–strain curves (**a**) and work-hardening curves (**b**) of different micro-alloyed Q&P steels.

Figure 6 shows the fracture morphology of the designed steels after the tensile test to clarify the fracture mechanism. Cracks, cleavages, and dimples were observed in different micro-alloyed steels. The cleavages and cracks were observed on the surface of V-alloyed steel (Figure 6a). The dimples and cleavages were found in Nb-Ti-alloyed steel (Figure 6b). In addition, the dimples, cleavages, and cracks appeared simultaneously in Ti-alloyed steel (Figure 6c). It can be easily seen that many more dimples were distributed in Nb-Ti-alloyed steel, and the cleavage size was much smaller than that of the other two steels. This case is consistent with the engineering stress–strain curves. It indicates that Nb-Ti-alloyed steel showed the best ductility, evidenced by the maximum necking value of Nb-Ti-alloyed steel obtained in the FE analysis relative to other steels, as shown in Figure 7.

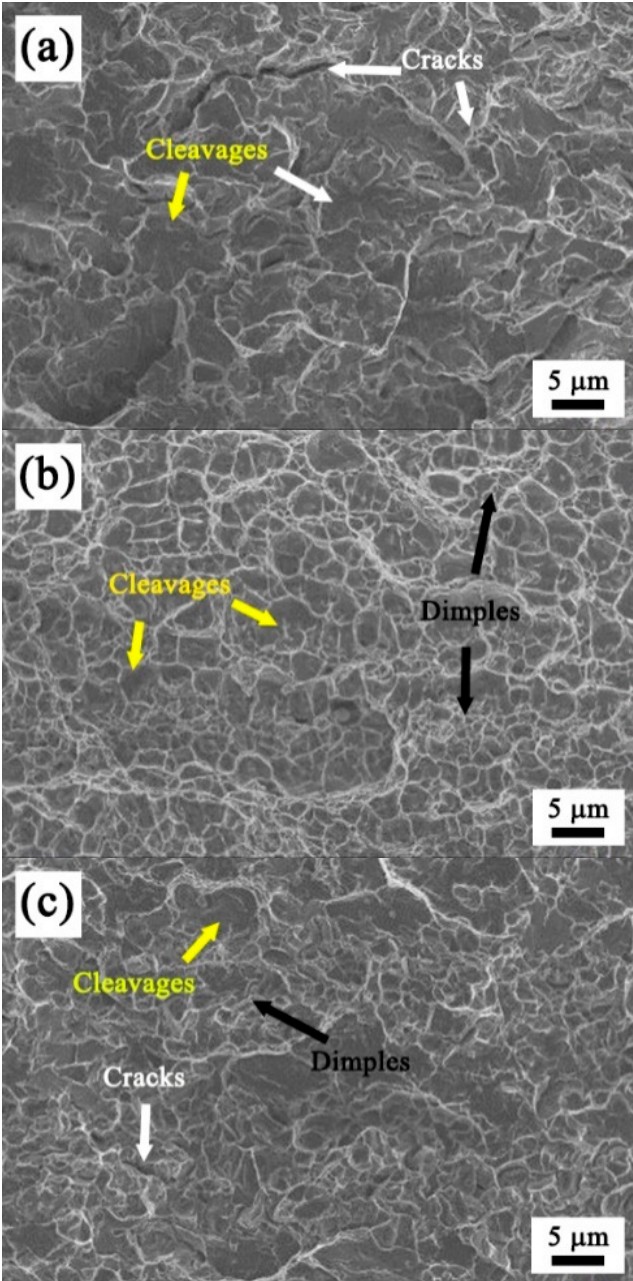

**Figure 6.** Fracture surface of (**a**) V-alloyed, (**b**) Nb-Ti-alloyed, and (**c**) Ti-alloyed steel.

In order to further verify the mechanical properties of three different micro-alloyed Q&P steels in terms of ductility, toughness, and plasticity, numerical simulations of tensile processes were respectively performed for these steels based on ABAQUS software. The model dimensions of the tensile specimens were $20 \times 1.5$ mm$^2$ in the cross-section and 80 mm in the gauge length along the transverse direction. The mesh size of the numerical model was around $2 \times 2$ mm, and the material properties were set according to the engineering stress–strain curves in Figure 5. Figure 7 shows the stress distribution and necking shrinkage of each micro-alloyed Q&P steel before fracture damage. The fracture stress of Nb-Ti-alloyed steel was the most minor compared with other steels, with a value of $1.111 \times 10^3$ MPa, and the size after necking was also the smallest compared with others, with a value of 9.32 mm, which means that the ductility of Nb-Ti-alloyed steel was the largest, from 20 mm necking to 9.32 mm. Thus, the numerical simulation analysis results confirm that Nb-Ti-alloyed steel had better ductility and plasticity.

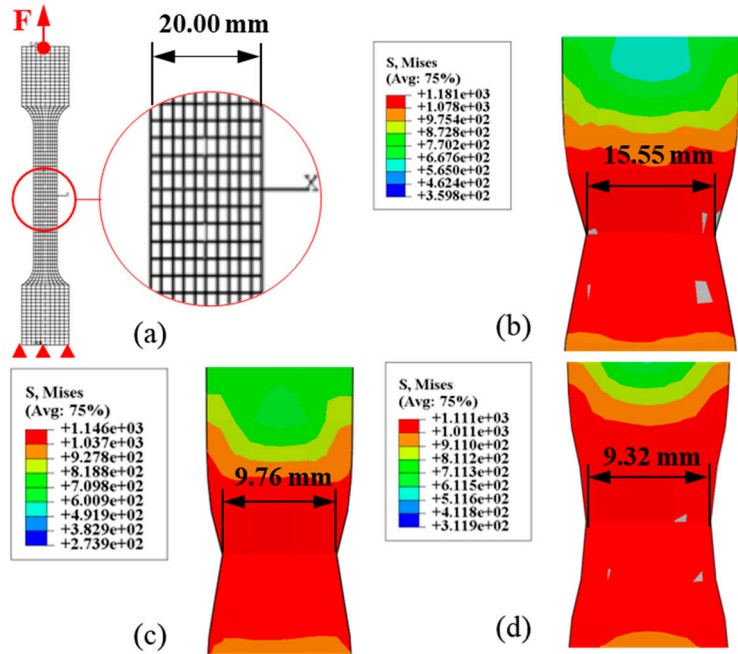

**Figure 7.** (**a**) The FE model for all steels, and the necking situations of (**b**) V-alloyed, (**c**) Ti-alloyed, and (**d**) Nb-Ti-alloyed steels.

## 4. Discussion

### 4.1. Stability of RA

It is well-known that the stability of RA is one of the main factors determining the mechanical properties of Q&P steel. The TRIP effect formula is given as [29]:

$$\log\left[ln\frac{f_s}{f_s - f}\right] = \log k + m \log \varepsilon \tag{2}$$

where, $f_s$ and $f$ refer to the volume fraction of the saturated martensite and the deformation-induced martensite (DIM), respectively, $\varepsilon$ represents the true strain, $m$ represents the coefficient of deformation mode, which keeps constant at 1.0, and $k$ is the coefficient of stability of RA, i.e., the larger the $k$ value, the lower the stability of RA.

As stated in Figure 4b, the RA volume fraction did not change as these designed steels were deformed within the range of elastic deformation. The change of volume fraction of RA was divided into two stages during the tensile tests. The change of RA volume fraction was not noticeable when the strain was relatively low. In contrast, the volume fraction of RA significantly decreased when the strain reached a certain level (Figure 4b) [30]. Therefore, the TRIP effect became remarkable after a true critical strain. Thus, Equation (2) is converted to Equation (3):

$$f/f_s = 1 - \exp(-k \times (\varepsilon - \varepsilon_0)) \tag{3}$$

According to the in situ analysis of RA, as shown in Figure 4, the experimental results were fitted very well using Equation (3) (Figure 8). Interestingly, the minimum value of $k$ among these three sheets of steel was 7.60, in V-alloyed steel. This case indicates that the stability of RA in V-alloyed steel was higher than that of the other steels, and the elongation of V-alloyed steel was lower. The $k$ value is an important index to evaluate the stability of RA. The fitting parameter obtained by using a statistical hypothesis test shows that the $k$ value of Ti-alloyed steel was higher than other steels, indicating that the RA stability in Ti-alloyed steel was weak. The RA in V-alloyed steel showed more substantial stability. The DIM transformation does not always enhance the plasticity when the austenite with low stability is large [31]. In the present study, most of the M/A islands of V-alloyed steel

were blocky, with an average size of ~1 μm (Figures 2d and 3a). The brittle fracture in V-alloyed steel could ascribe to the blocky microstructures, since the blocky RA generally transforms to martensite at an earlier stage than the thin-film RA [32,33], which promotes the initiation of brittle cracking. This phenomenon is also confirmed by another study [2]. The transformation of metastable RA effectively enhanced the fracture resistance of Q&P steel. As the stress triaxiality varied, the transformation of RA with 8 vol % was proven to absorb 22–35% of the increments of the work of fracture [3].

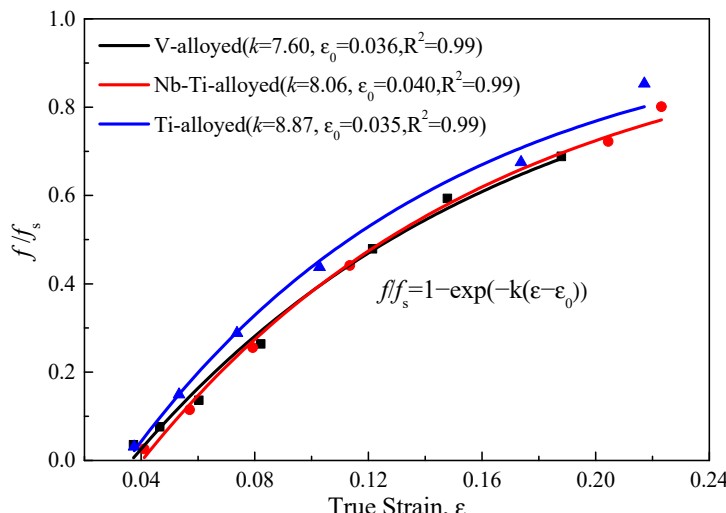

**Figure 8.** Relationship between martensitic transformation and true strain.

### 4.2. Strengthening Mechanism

Yield strength for the investigated steels showed an apparent difference. Compared with the yield strength of V-alloyed steel (668 MPa) and Nb-Ti-alloyed steel (622 MPa) obtained by the one-step Q&P process, the yield strength of Ti-alloyed (718 MPa) steel subjected to the two-step Q&P process was higher. In multi-phase steels, the yield strength is determined by the soft phase. Assisted by the neutron diffraction pattern in TRIP-assisted steel [33–35], ferrite is the preferential yield phase during the tensile test process. The yield strength was evaluated by the following equation [36]:

$$\sigma_y = \sigma_0 + \Delta\sigma_S + \Delta\sigma_G + \Delta\sigma_d + \Delta\sigma_p \tag{4}$$

where $\sigma_y$ is the predicted yield strength, $\sigma_0$ is the lattice friction stress, $\Delta\sigma_s$ are the strengthening increments caused by solid solution, and $\Delta\sigma_G$, $\Delta\sigma_d$, and $\Delta\sigma_p$ are the increments caused by sufficient ferrite grain size, dislocation, and precipitation, respectively.

In the present study, the lattice friction was:

$$\sigma_0 = 48 \text{ MPa} \tag{5}$$

The effect of the solid solution strengthening is provided as follows [37,38]:

$$\Delta\sigma_S = 4750[C] + 3750[N] + 37[Mn] + 84[Si] \tag{6}$$

where [i] is the element's content in solid solution in ferrite, in mass %.

There was no yield platform during the tensile test, and the carbonitrides were precipitated in ferrite. Thus, the role of solid carbon and nitrogen atoms can be ignored. The values of solid solution strengthening of V-alloyed, Nb-Ti-alloyed, and Ti-alloyed steels were calculated for 213 MPa, 219 MPa, and 235 MPa, respectively. The effect of the grain size strengthening was calculated via Equation (7):

$$\Delta\sigma_G = k_y d^{-1/2} \tag{7}$$

where, $k_y = 17.4 \text{ MPa·mm}^{1/2}$, and $d$ is the average ferrite grain size in mm.

According to the results of the average grain size of ferrite in V-alloyed, Nb-Ti-alloyed, and Ti-alloyed steels (Figure 2), the hardening values of the grain size were 216 MPa, 190 MPa, and 258 MPa, respectively. The effect of the dislocation strengthening was obtained via Equation (8):

$$\Delta\sigma_d = \alpha G b \rho^{1/2} \tag{8}$$

where $\alpha$ is a numerical factor dependent on the crystal structure, $G$ is the shear modulus, $8.3 \times 10^4$ MPa, $b$ is the Burgers vector of a dislocation, 0.248 nm, and $\rho$ is the dislocation density, $1/m^2$. It is reported that the dislocation density in ferrite is in the order of $10^9$ to $10^{10}$ cm$^{-2}$ [39].

If the dislocation density in ferrite is assumed as $10^9$ cm$^{-2}$, the value of the dislocation hardening is about 25 MPa. Additionally, the effect of the precipitation strengthening was calculated by Equation (9) [40]:

$$\Delta\sigma_p = \frac{0.538 G b f^{\frac{1}{2}}}{X} ln\left(\frac{X}{2b}\right) \tag{9}$$

where, $G$ is the shear modulus, $b$ is the Burger vector, $f$ is the volume fraction of precipitates, and $X$ is the diameter of the precipitates.

The values of precipitation hardening were calculated as 157 MPa, 135 MPa, and 89 MPa for V-alloyed, Nb-Ti-alloyed, and Ti-alloyed steels, respectively. Figure 9 shows the calculated and experimental yield strength. The calculated values were very close to the experimental ones in all three steel sheets. In contrast, the calculated yield strength of Ti-alloyed steel was lower than that of the experimental value (the difference was 54 MPa). This may be mainly due to the following two factors. On the one hand, the dislocation density was insufficiently estimated for the two-step Q&P process. On the other hand, the carbon-trapping sites at dislocations and grain boundaries of ferrite drastically increased the yield strength after the partitioning process [41], such as a 50~150 MPa improvement in yield strength by the tempering process of DP steels [42], and an over 300 MPa yield strength improvement in ferrite-bearing Q&P steels [41].

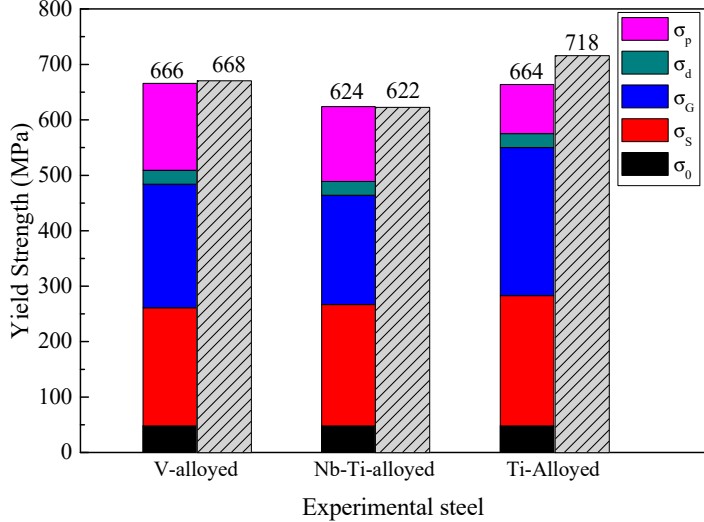

**Figure 9.** The comparison of calculated and experimental yield strength in the experimental steels.

According to the compound strengthening, the estimate of the flow stress of multiphase TRIP steels [43,44] was calculated by Equation (10):

$$\sigma = f_\alpha \sigma_\alpha + f_\gamma \sigma_\gamma + f_m \sigma_m + f_B \sigma_B \tag{10}$$

When the true strain was more than 0.04, the RA was transformed to martensite. Assuming that the plastic deformation mechanism was mainly contributed by martensite transformation, the stress increment was calculated by:

$$\Delta\sigma = f(\sigma_m - \sigma_\gamma) \tag{11}$$

The above result indicates that the stress was linear to the volume fraction of martensite transformation. Figure 10 shows a linear fitting curve between the true strain and the experimental data. The stress increments for 1 vol % RA transformation during the tensile deformation were 30.38 MPa (Ti-alloyed), 31.64 MPa (Nb-Ti-alloyed), and 46.37 MPa (V-alloyed), respectively. Noticeably, the stress increment of V-alloyed steel was higher than that of Nb-Ti-alloyed and Ti-alloyed steels. Moreover, the strength difference between martensite and austenite in V-alloyed steel was higher than that of Nb-Ti-alloyed and Ti-alloyed steels.

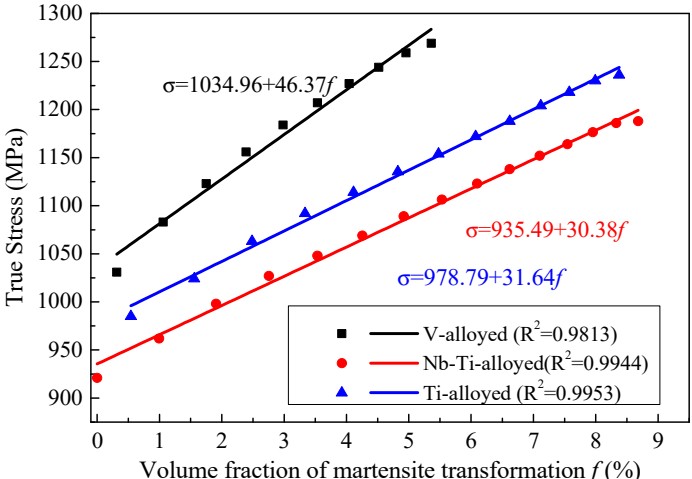

**Figure 10.** Relationship between true stress and volume fraction of martensite transformation.

The value of the work-hardening rate (Figure 5b) is an important index to evaluate the TRIP effect. The work-hardening rate for the V-alloyed specimens showed a definite downward trend. For Nb-Ti-alloyed and Ti-alloyed samples, the work-hardening rate showed three stages (S1, S2, and S3, respectively (Figure 5b)): sharp decline, a slight rise, and a slow decline. Dislocation slip in ferrite and martensite led to a sharp decline in the work-hardening rate. With the increase of the true strain, the volume of retained austenite slowly increased, and the TRIP effect occurred. The retained austenite was transformed into $\alpha'$ martensite under stress, causing the work-hardening rate to be accelerated. The transformed $\alpha'$ martensite suppressed the nucleation and growth of micro-voids and microcracks. Thus, a local area of strengthening was formed.

### 4.3. Effects of Alloying Elements on Microstructures and Properties

Nb, V, and Ti elements can improve the strength and ductility for high-strength steels through grain refinement and precipitation. Furthermore, solid carbide-forming elements will be combined with carbon atoms in the austenite region. They result in less stability of austenite. The investigated 1000 MPa-grade, advanced, high-strength steel with products of tensile strength and elongation of 17.84 GPa·% to 24.12 GPa·% were obtained by V-alloyed, Nb-Ti-alloyed, and Ti-alloyed steels, respectively. The excellent combination of high strength and toughness was superior to Nb-alloyed Q&P steels with 15.72 GPa·% to 18.47 GPa·% [10,45] and Nb-V-alloyed and Ti-alloyed TRIP steels with 17.27 GPa·% to 21.41 GPa·% [7,8,46–49]. It is essential to control the precipitation to balance tensile strength and ductility in micro-alloyed steels. For the investigated V-alloyed steel, the precipitating and coarsening of VC did not weaken the stability of austenite. Its

microstructure was mostly similar to ferrite and martensite dual-phase steels. In Nb-Ti-alloyed and Ti-alloyed steels, the precipitates were below 20 nm and had a negligible effect on the carbon concentration of RA. The carbon atoms' diffusion process was from supersaturated martensite to RA, causing most of the retained austenite to be embedded by the initial martensite. However, the transformation of martensite and bainite was challenging due to an insufficient driving force, such as quenching and partitioning near $M_s$. As a result, the carbon atoms partitioned inner untransformed austenite. Therefore, blocky-retained austenite and martensite were mainly observed in the one-step Q&P processes. In addition, another discovery from the current work is vanadium, which is known for providing precipitation reinforcement of steel and has generated much interest over the past half-century. However, the effect of vanadium in Q&P steel is not notably reported. In Q&P steel studied in this paper, vanadium carbide (VC) precipitation occurred at the interface of migrating austenite/ferrite, i.e., the interface phase precipitation, and occurred in the ferrite. The solubility of vanadium carbide in austenite was higher than that of Ti and Nb carbide, so it is not easy to form vanadium carbide in austenite. The decrease of the solubility of carbide in the process of austenite to ferrite is conducive to the formation of phase precipitation and ferrite, which reduces the coarsening rate of the precipitated phase and leads to its finer distribution, which is very important for the hardening of steel. The beneficial contribution of vanadium carbide to the overall mechanical properties of steel and the necessity of optimizing vanadium utilization require more studies on vanadium carbide precipitation and its interaction with austenitic, ferritic transformation.

## 5. Conclusions

This investigation compared the microstructure of Q&P steels with different compositions prepared by different Q&P processes. Through the in situ tensile test and X-ray technology, the relationship between the ductility of Q&P steel and the size and morphology of RA was described. The following main conclusions can be presented:

1. The following equation well-described the strain-induced martensite transformation of RA: $f/f_s = 1 - \exp(-k(\varepsilon - \varepsilon^0))$. The TRIP effect became the primary deformation mechanism as the strain overcame 0.04.

2. The calculated yield strength was fitted very well with the experimental value for the one-step Q&P steel. A slight deviation was observed in the two-step Q&P steel, mainly due to the incomplete consideration of dislocation density and carbon trapping in the two-step Q&P process.

3. The true stress maintained a linear relationship with the volume fraction of martensite transformation. Furthermore, the stress increments for 1 vol % RA transformed to martensite during the tensile deformation were 30.38 MPa (Ti-alloyed), 31.64 MPa (Nb-Ti-alloyed), and 46.37 MPa (V-alloyed), respectively.

4. The product of tensile strength and elongation of 24.12 GPa·% was obtained in Nb-Ti-alloyed steel after being subjected to a one- or two-step Q&P process.

**Author Contributions:** Conceptualization, Z.C.; methodology, X.Y. and X.G.; software, Z.C. and X.G.; formal analysis, H.Z.; investigation, X.Y. and H.Z.; resources, G.Z.; data curation, Z.Z. and G.S.; writing—original draft, X.Y. and G.Z.; writing—review and editing, Z.H. and G.S.; visualization, Z.Z.; supervision, Z.H.; project administration, G.Z. All authors have read and agreed to the published version of the manuscript.

**Funding:** The authors gratefully acknowledge financial support from High-tech and Industrialization Projects (No. 2016CY-G-13 and No. 2017CY-G-3) for Panzhihua Government of Sichuan Province, China.

**Data Availability Statement:** Not applicable.

**Conflicts of Interest:** The authors declare no conflict of interest.

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
