# Peer review of "In Situ Observation of Retained Austenite Transformation in Low-Carbon Micro-Alloyed Q&P Steels"

_crystals, doi:10.3390/cryst13020351_

Round 1

Reviewer 1 Report

The presented study entitled In-situ observation of retained austenite transformation in low carbon micro-alloyed Q&P steels represents a rather interesting study. I very much appreciate the experimental verification carried out. The study is logically laid out with fairly illustrative pictures and tables. The obtained results are duly discussed. Despite the positive feeling about the presented study, I have several comments and questions for the authors. 1. In the study, despite the definition of the main goal and procedure, I lack confrontation with other studies that deal with the given issue. Without this, it is very difficult to identify the contribution, the novelty of the given study. Here I will ask the authors to supplement and incorporate it into the introductory part of the study. At the same time, I would welcome exact definitions of the contribution and novelty of the study. 2. It would be good to expand the abstract to include the conditions of experimental verification and at the same time to expand it by defining the main results. 3. Despite the fact that the authors made repeated measurements, a basic statistical analysis of the achieved results is missing. I would welcome a comparison of the basic results using statistical hypothesis tests to really confirm the differences in the studied parameters between the studied steels. 4. Fig.8 - what does the value of R2 represent? I miss her interpretation in the text. Fill in please. 5. Despite the interesting results, the conclusion is relatively meager. I will ask the authors to expand the conclusion, for possible practical implications of the results for practical use. 6. Please complete the limitations of the research and results.

Author Response

Thank you for giving us a chance to revise the manuscript again. We have substantially revised our manuscript after reading the comments provided by the reviewer and editor.

We have read the reviewer's comments carefully and have made revisions, which were highlighted with yellow in the revised manuscript. We want to express our great appreciation to the editor and reviewers for comments on our manuscript.

Thank you and best regards.

Reviewer 2 Report

Journal: Crystals

Manuscript ID: crystals-2202229

Title: In-situ observation of retained austenite transformation in low carbon micro-alloyed Q&P steels

Authors: Xiaoyu Ye, Haoqing Zheng, Gongting Zhang, Zhiyuan Chang, Zhiwang Zheng, Zhenyi Huang, Xiuhua Gao, Guanqiao Su

This work investigates the structure of low-carbon Micro-Alloyed steels during strain. It is significant that the study of retained austenite is made in-situ, it is very interesting and relevant. Although the obtained dependence is almost linear, it is still an important result. It is also very interesting to calculate the contributions of different hardening mechanisms, which in general is close to the experimental values. The work is well done and of interest to the journal audience. The remarks are mostly formal in nature:

1.      I would like to see a diagram and/or photo of In-Situ Observation.

2.      Figure 2: the inscriptions in Figures 2a, 2b, 2c are very poorly visible.

3.      Table 2: it does not make sense to specify 2 decimal places if the error of measurement is integer. It would be more correct to indicate 17±2 and 10.8±0.7. You should also indicate the units of measurement.

4.      Figure 3: there is no scale in Figures 3a and 3b. Also it is necessary to indicate on figures retained austenite and carbides.

5.      Figure 4: some information is lost between the peaks in the diffractogram.

6.      Figure 7: incomplete description of the figure. The FE model of what? Are these stresses or strains? In what units?

7.      The motivation for the simulation is completely unclear. In addition, the results of the simulation did not provide any information. In fact, the results are described in one sentence, from which it follows that the Nb-Ti specimens had the highest ductility. However, this was known from mechanical testing. The neck would have been better measured on real specimens. All in all, the whole modeling part raises big questions.

Reviewer 3 Report

In this article, the authors studied the in-situ analysis of RA and tracked its volume fraction during the tensile tests to investigate their stability of in the micro-alloyed low carbon Q&P steels, and the associated strengthening mechanisms. They show that dislocation density and carbon trapping are important in estimation of the yield strengths of the two-step Q&P steels.

While the study is interesting, the authors are requested to do minor revision by making the following corrections/changes:

In line 62, it will be great if the authors could please provide some reference to the studies that shows significant advantage in the in-situ analysis of RA using X-ray stress apparatus.

How were the chemical compositions of the Q&P steels determined? Its not mentioned in the article.

What was the purpose of cooling the samples to 760C after 820C?

Scale bars in 2a and 2b. Mention which figures are from optical and which are from SEM in the caption or in texts. Please use different color arrows/texts in 2b and 2c for clear representation.

Line 137: please mention the full form of M/A abbreviation here instead of line 140 since it was used for the first time here. 

In this study, the initial microstructures of the alloys studied are not shown, i.e. the microstructures after solutionizing but before one or two-step quenching processes. It will be better to show them in order to clarify of they were solutionized properly and there is no effect of the parent microstructures on the following heat treatments.

In figure 3, please mark the ferrite, RA, carbides, etc in the figure to make it clear. Please mention if these are bright field TEM images/STEM images.

Line 167: the authors mention that (Nb, Ti)C and TiC are primary type of precipitation in V-alloyed steel. Please explain, since there is no Nb and Ti in this particular alloy. I think VC was missing in the above statement.

In Figure 4, please make the color codings in figures 4a and 4b consistent (also with figure 5, figure 8 and figure 10).

Line 193: authors claim that the Nb-Ti alloy shows the best stability of RA. The statement provided here doesn't seem to clearly justify this claim.

Under section 3.3, figure 6c is not talked about at all. please mention something about it.

Please provide some explanation of figure 7 under section 3.

Thanks!

Reviewer 4 Report

This paper is meaningful and suitable for publication, but several points should be considered by the authors prior to publication.

1. In the last paragraph of the introduction section, the authors should claim the aim of this paper. 

2. For the results shown in Fig.2, I suggested the authors could compare the difference in microstructure morphology of the one-step Q&P process and two-step Q&P.

3. The authors should add a discussion about the relationship between observed microstructure and mechanical properties.

4. In the abstract section, I hope the author could point out the role of alloying elements on microstructure morphology or mechanical properties.

5. Please correct the reference format and the format should be uniform.

Reviewer 5 Report

This manuscript has too many similarities with Reference [21], where the same author of Pangang Group is co-authored. Therefore, the differences between the two works and the novelty of the present work over another should be clearly explained. Otherwise, I do not believe that the results of the present study may add significant findings and novelty to the pre-existing literature results. In addition, I have a few more comments for the authors:

1- The Abstract should be rewritten, focusing on the main findings. At this stage, the Abstract is not meaningful.

2- Line 62-64: Recently, the X-ray stress apparatus (X-350), aided by a micro-electronic universal testing machine, shows …’. Is this the finding of another work or the present work? In the case of the former, it seems that the citation is missing.

3- The layout of Figure 2 is confusing. Use a-b for V-alloyed, c-d for Nb-Ti steel, and e-f for Ti-steel.

4- To characterize precipitates, it is better to show the TEM diffraction patterns as well.

5- More data on FE modeling by Abaqus is required.

6- How did the authors obtain the lattice fraction as 45 MPa in Equation 5?

7- Equation 6 is missing a citation.

8- Line 171-175, why are volume fractions of carbides reduced? It should be clearly explained.

9- In the Introduction, the authors mentioned that the RA transformation and strengthening mechanism is unclear in the Q&P steels in which the vital carbide-forming elements Nb, V, and Ti are added. However, the manuscript is missing a detailed discussion on carbide formation and (possible) dissolution due to HT.

Round 2

Reviewer 5 Report

The authors sufficiently addressed my accounts. I recommend the paper to publish in Crystals.